# Embarrassingly Simple Dataset Distillation

**Yunzhen Feng** [*][‡]   **Ramakrishna Vedantam** [*]   **Julia Kempe** [*][†]
[*] Center for Data Science, New York University
[†] Courant Institue of Mathematical Sciences, New York University
[‡] yf2231@nyu.edu

## Abstract

Training of large-scale models in general requires enormous amounts of traning data. Dataset distillation aims to extract a small set of synthetic training samples from a large dataset with the goal of achieving competitive performance on test data when trained on this sample, thus reducing both dataset size and training time. In this work, we tackle dataset distillation at its core by treating it directly as a bilevel optimization problem. Re-examining the foundational back-propagation through time method, we study the pronounced variance in the gradients, computational burden, and long-term dependencies. We introduce an improved method: Random Truncated Backpropagation Through Time (RaT-BPTT) to address them. RaT-BPTT incorporates a truncation coupled with a random window, effectively stabilizing the gradients and speeding up the optimization while covering long dependencies. This allows us to establish new dataset distillation state-of-the-art for a variety of standard dataset benchmarks.

## 1   Introduction

*Dataset Distillation*, introduced by [54], aims to condense a given dataset into a small synthetic version such that when neural networks are trained on this distilled version, they achieve good performance on the original distribution. The distilled datasets thus speed up model-training [36] by using less data and training steps, and have found numerous applications including protecting privacy [11, 5], continual learning [55, 42], federated learning [59, 29], and neural architecture search [47].

Dataset distillation is an instance of bilevel optimization [8] where one optimization output (in this instance, the learning algorithm trained on the small dataset) is fed into another optimization problem (the generalization error on the target set) which we intend to minimize. In general, this problem is intractable, as the inner loop involves a multi-step computation with a large number of steps. Early works [54, 48, 9] tackle this problem via back-propagation through time (BPTT), the go-to method for bilevel optimization [14, 34]. BPTT unrolls the inner loop for a certain number of steps and calculate the meta-gradient for the distilled dataset. However, long unrolling introduces large computational and memory requirements, limiting performance. Numerous follow-up works turn to replacing the inner loop with closed-form differentiable *surrogates* [37, 38, 64, 32] or modify the outer loop objective using *proxy training-metrics* [3, 62, 63] (see the appendix for related work).

In this paper, we refine BPTT and achieve state-of-the-art performance across a vast majority of the CIFAR10, CIFAR100, CUB and TinyImageNet benchmarks. For dataset distillation, the inner problem presents unique challenges – the pronounced non-convex nature when training a neural network from scratch on the distilled data. One has to use long unrolling to encapsulate the long dependencies inherent in the inner optimization. However, this results in BPTT suffering from slow optimization and huge memory demands, a consequence of backpropagating through all intermediate steps. This is further complicated by considerable instability in meta-gradients, emerging from the multiplication of Hessian matrices during long unrolling. Therefore, the performance is limited.

---

Workshop on Advancing Neural Network Training at 37th Conference on Neural Information Processing Systems (WANT@NeurIPS 2023).

To address these challenges, we integrate the concepts of randomization and truncation with BPTT, leading to the Random Truncated Backpropagation Through Time (RaT-BPTT) method. The refined approach unrolls within a randomly anchored smaller fixed-size window along the training trajectory and aggregates gradients within that window (see Fig. 1 for a cartoon illustration). The random window design ensures that the RaT-BPTT gradient serves as a random subsample of the full BPTT gradient, covering the entire trajectory, while the truncated window design enhances gradient stability and alleviates memory burden. Consequently, RaT-BPTT provides expedited training and superior performance compared to BPTT.

Overall, our method is *embarrassingly* simple – we show that a careful analysis and modification of backpropagation lead to results exceeding the current state-of-the-art, without resorting to various approximations, a pool of models in the optimization, or additional heuristics. Since our approach does not depend on large-width approximations, it works for any architecture, in particular commonly used narrower models, for which methods that use inner-loop approximations perform less well. Moreover, our method can be seamlessly combined with prior methods on dataset re-parameterization [9], leading to further improvements. To our knowledge, we are the first to introduce *truncated* backpropagation through time [44] to the dataset distillation setting, and to combine it with *random* positioning of the unrolling window.

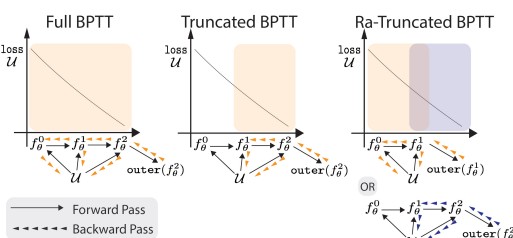

Figure 1: **Illustration of bilevel optimization** of the `outer` loss when training for 2 steps. We show BPTT (left), Truncated BPTT (middle) and our proposed RaT-BPTT (right). RaT-BPTT picks a window in the learning trajectory (randomly) and tracks the gradients for the chosen window, as opposed to T-BPTT that uses a fixed window, and BPTT that uses the entire trajectory.

## 2 Methods

Denote the original training set as $\mathcal{D}$ and the distilled set as $\mathcal{U}$. With an initialization $\theta_0$ for the inner-loop learner $\mathcal{A}$, we perform the optimization for $T$ steps to obtain $\theta_T(\mathcal{U})$ with loss $\mathcal{L}(\theta_T(\mathcal{U}), \mathcal{D})$. We add $(\mathcal{U})$ to denote its dependence on $\mathcal{U}$. The dataset distillation problem can be formulated as

$$\min_{\mathcal{U}} \ \mathcal{L}(\theta_T(\mathcal{U}), \mathcal{D}) \ \text{(outer loop)} \quad \text{such that} \quad \theta_T(\mathcal{U}) = \mathcal{A}(\theta_0, \mathcal{U}, T) \ \text{(inner loop)} \tag{1}$$

When the inner-loop learner $\mathcal{A}$ is gradient descent with learning rate $\alpha$, one could leverage the chain rule to get the gradient of BPTT with respect to the distilled data:

$$\mathcal{G}_{BPTT} = -\alpha \frac{\partial \mathcal{L}(\theta_T(\mathcal{U}), \mathcal{D})}{\partial \theta} \sum_{i=1}^{T-1} \Pi_{j=i+1}^{T-1} \left[ 1 - \alpha \frac{\partial^2 \mathcal{L}(\theta_j(\mathcal{U}), \mathcal{U})}{\partial \theta^2} \right] \frac{\partial^2 \mathcal{L}(\theta_i(\mathcal{U}), \mathcal{U})}{\partial \theta \partial U} \tag{2}$$

This computation indicates that the meta-gradient is divided into $T-1$ segments. Each part represents a matrix product $\Pi[1 - \alpha H]$ where each $H$ matrix corresponds to a Hessian matrix. Yet, computing the meta-gradient demands the storage of all intermediate states to backpropagate through, and thus is less computationally efficient.

To circumvent these challenges, the prevalent strategy is to adopt the *truncated* BPTT (T-BPTT) method [56, 39], which unrolls the inner loop for the same $T$ steps but only propagates backwards through a smaller window of $M$ steps. Therefore, in the T-BPTT gradient, the sum in Eq. (2) starts at $T - M$. This technique omits the initial $T - M - 1$ terms, each being a product of more than $M$ Hessian matrices. Assuming the inner loss function is strongly convex, T-BPTT aligns well with BPTT [44]. The convexity assumption implies positive eigenvalues of the Hessians, causing the term $\Pi[1 - \alpha H]$ to vanish as the number of factors increases, allowing for good performance of T-BPTT with less memory requirement and faster optimization time. However, in our scenario, the task involves training a random neural network on distilled data; it is thus inherently non-convex, with multiple minima.

We visualize the training curve and the norm of meta-gradients through outer-loop optimization steps in Fig. 3 and Fig. 2, respectively. All experiments are on CIFAR10 with IPC (image per class) 10. A comparison between BPTT and T-BPTT reveals that: 1) meta-gradients from T-BPTT show more

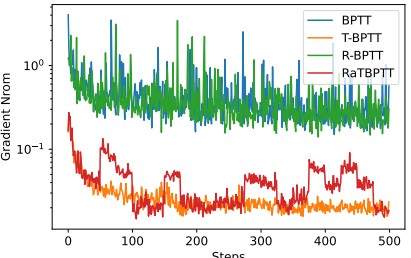

Figure 2: Meta-gradient norm in the first 500 steps. BPTT: unroll 120 steps). T-BPTT: unroll 120 steps and backpropagate 40 steps. RaT-BPTT: we randomly place the backpropagation window for each epoch (25 steps)

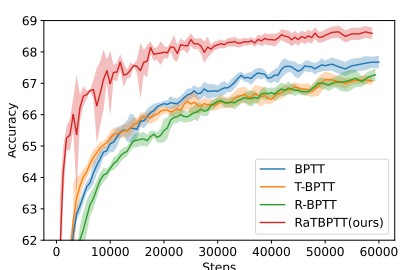

Figure 3: Test Accuracy during distillation with BPTT, T-BPTT, R-BPTT, and our RaT-BPTT.

stability than from BPTT, largely due to the excluded $T - M + 1$ gradient terms. This highlights the non-convexity of the inner problem, marked by Hessian matrices having negative eigenvalues. The impact of these eigenvalues intensifies the variance, resulting in unstable gradients. However, once gradients stabilize, T-BPTT displays swift early progress. 2) BPTT ends up with higher accuracy than T-BPTT, suggesting T-BPTT might overlook vital initial phase details, crucial since neural network optimization often peaks early in the inner loop. The challenge thus is how to merge the good performance of BPTT with the computational speedup of T-BPTT.

To this end, we propose the *Random* Truncated BPTT (RaT-BPTT) in Alg. 1, which randomly places the truncated window along the inner unrolling chain. The gradient of RaT-BPTT is

$$\mathcal{G}_{RaT-BPTT} = -\alpha \frac{\partial \mathcal{L}(\theta_N(\mathcal{U}), \mathcal{D})}{\partial \theta} \sum_{i=N-M}^{N-1} \Pi_{j=i+1}^{N-1} \left[ 1 - \alpha \frac{\partial^2 \mathcal{L}(\theta_j(\mathcal{U}), \mathcal{U})}{\partial \theta^2} \right] \frac{\partial^2 \mathcal{L}(\theta_i(\mathcal{U}), \mathcal{U})}{\partial \theta \partial U} \quad (3)$$

Looking at the gradients, RaT-BPTT differs by randomly sampling M consecutive parts in $\mathcal{G}_{BPTT}$ and leaving out the shared Hessian matrix products. Therefore, RaT-BPTT is a subsample version of BPTT, spanning the entire learning trajectory. Moreover, the maximum number of Hessians in the product is restricted to less than M. It thus inherits the benefits of both the accelerated performance and gradient stabilization from T-BPTT. As illustrated in Fig. 3, RaT-BPTT consistently outperforms other methods throughout the optimization process. We also examine performing full unrolling along trajectories of randomly sampled lengths (R-BPTT) as a sanity check. The gradients are similarly unstable and the performance is worse than full unrolling with BPTT.

## 3 Experimental Results

In this section, we present an evaluation of our method, RaT-BPTT, comparing it to a range of SOTA methods across multiple benchmark datasets.

**Datasets** We run experiments on four standard datasets, CIFAR-10 (10 classes, $32 \times 32$), CIFAR-100 (100 classes, $32 \times 32$, [23]), Caltech Birds 2011 (200 classes, CUB200, $32 \times 32$, [51]) and Tiny-ImageNet (200 classes, $64 \times 64$, [24] ). We distill datasets with 1, 10, and 50 images per class for the first two datasets and with 1 and 10 images per class for the last two datasets.

**Baselines** We compare our methods to two lines of SOTA methods 1) *Inner-loop surrogates*: BPTT (the non-factorized version of LinBa in [9]), Neural Tangent Kernel (KIP) [38], Random Gaussian Process (RFAD) [32], and empirical feature kernel (FRePO) [64], and reparameterized convex implicit gradient (RCIG)

---

**Algorithm 1** Dataset Distillation with RaT-BPTT. Differences from BPTT are highlighted in purple.

**Input:** Target dataset $\mathcal{D}$. N: total number of unrolling steps. T: truncated window size.

1: Initialize distilled data $\mathcal{U}$ from Gaussian
2: **while** Not converged **do**
3:     Uniformly sample M in $[0, N-T]$ as the current unrolling length
4:     Sample a batch of data $d \sim \mathcal{D}$
5:     Randomly initialize $\theta_0$ from $p(\theta)$
6:     **for** $t = 0 \rightarrow M + T - 1$ **do**
7:         If $t == M$, start accumulating gradients
8:         Sample a mini-batch of distilled data $u_t \sim \mathcal{U}$
9:         Update network $\theta_{t+1} = \theta_t - \alpha \nabla \ell(u_t; \theta_t)$
10:     **end for**
11:     Compute classification loss $\mathcal{L} = \ell(d, \theta_{T+M})$
12:     Update $\mathcal{U}$ with respect to $\mathcal{L}$.
13: **end while**

Table 1: Performance on standard datasets: The **AVG** column shows the average performance across datasets. * indicates evaluations using wider ConvNets. FRePO and RCIG are the re-evaluated results with narrow networks. Results highlight top performance for narrow networks while results are best for wide networks.

| Dataset | CIFAR-10 | | | CIFAR-100 | | | CUB200 | | T-ImageNet | | AVG |
| Img/class(IPC) | 1 | 10 | 50 | 1 | 10 | 50 | 1 | 10 | 1 | 10 | |
|---|---|---|---|---|---|---|---|---|---|---|---|
| **Inner Loop** | | | | | | | | | | | |
| BPTT [9] | $49.1_{\pm0.6}$ | $62.4_{\pm0.4}$ | $70.5_{\pm0.4}$ | $21.3_{\pm0.6}$ | $34.7_{\pm0.5}$ | - | - | - | - | - | - |
| KIP* [38] | $49.9_{\pm0.2}$ | $62.7_{\pm0.3}$ | $68.6_{\pm0.2}$ | $15.7_{\pm0.2}$ | $28.3_{\pm0.1}$ | - | - | - | - | - | - |
| RFAD* [32] | $53.6_{\pm1.2}$ | $66.3_{\pm0.5}$ | $71.1_{\pm0.4}$ | $26.3_{\pm1.1}$ | $33.0_{\pm0.3}$ | - | - | - | - | - | - |
| FRePO* [64] | $46.8_{\pm0.7}$ | $65.5_{\pm0.6}$ | $71.7_{\pm0.2}$ | $28.7_{\pm0.1}$ | $42.5_{\pm0.2}$ | $44.3_{\pm0.2}$ | $12.4_{\pm0.2}$ | $16.8_{\pm0.1}$ | $15.4_{\pm0.3}$ | $25.4_{\pm0.2}$ | $36.9_{\pm0.3}$ |
| FRePO | $45.6_{\pm0.1}$ | $63.5_{\pm0.1}$ | $70.7_{\pm0.1}$ | $26.3_{\pm0.1}$ | $41.3_{\pm0.1}$ | $41.5_{\pm0.1}$ | - | - | $16.9_{\pm0.1}$ | $22.4_{\pm0.1}$ | - |
| RCIG* [33] | $53.9_{\pm1.0}$ | $69.1_{\pm0.4}$ | $73.5_{\pm0.3}$ | $39.3_{\pm0.4}$ | $44.1_{\pm0.4}$ | $46.7_{\pm0.1}$ | $12.1_{\pm0.2}$ | $15.7_{\pm0.3}$ | $25.6_{\pm0.3}$ | $29.4_{\pm0.2}$ | $40.9_{\pm0.4}$ |
| RCIG | $49.6_{\pm1.2}$ | $66.8_{\pm0.3}$ | - | $35.5_{\pm0.7}$ | - | - | - | - | $22.4_{\pm0.3}$ | - | - |
| **Modified Objectives** | | | | | | | | | | | |
| DSA [61] | $28.8_{\pm0.7}$ | $52.1_{\pm0.5}$ | $60.6_{\pm0.5}$ | $13.9_{\pm0.3}$ | $32.3_{\pm0.3}$ | $42.8_{\pm0.4}$ | $1.3_{\pm0.1}$ | $4.5_{\pm0.3}$ | $6.6_{\pm0.2}$ | $14.4_{\pm0.2}$ | $25.7_{\pm0.7}$ |
| DM [63] | $26.0_{\pm0.8}$ | $48.9_{\pm0.6}$ | $63.0_{\pm0.4}$ | $11.4_{\pm0.3}$ | $29.7_{\pm0.3}$ | $43.6_{\pm0.4}$ | $1.6_{\pm0.1}$ | $4.4_{\pm0.2}$ | $3.9_{\pm0.2}$ | $12.9_{\pm0.4}$ | $24.5_{\pm0.4}$ |
| MTT [3] | $46.3_{\pm0.8}$ | $65.3_{\pm0.7}$ | $71.6_{\pm0.2}$ | $24.3_{\pm0.3}$ | $40.1_{\pm0.4}$ | $47.7_{\pm0.3}$ | $2.2_{\pm0.1}$ | - | $8.8_{\pm0.3}$ | $23.2_{\pm0.2}$ | - |
| FTD [12] | $46.8_{\pm0.3}$ | $66.6_{\pm0.3}$ | $73.8_{\pm0.2}$ | $25.2_{\pm0.2}$ | $43.4_{\pm0.3}$ | $50.7_{\pm0.3}$ | - | - | $10.4_{\pm0.3}$ | $24.5_{\pm0.2}$ | - |
| Ours | $53.2_{\pm0.7}$ | $69.4_{\pm0.4}$ | $75.3_{\pm0.3}$ | $35.3_{\pm0.4}$ | $47.5_{\pm0.2}$ | $50.6_{\pm0.2}$ | $13.8_{\pm0.3}$ | $17.7_{\pm0.2}$ | $20.1_{\pm0.3}$ | $24.4_{\pm0.2}$ | $40.8_{\pm0.3}$ |
| Ours (transfer to wide) | $54.1_{\pm0.4}$ | $71.0_{\pm0.2}$ | $75.4_{\pm0.2}$ | $36.5_{\pm0.3}$ | $47.9_{\pm0.2}$ | $51.0_{\pm0.3}$ | $14.2_{\pm0.3}$ | $17.9_{\pm0.3}$ | $20.3_{\pm0.1}$ | $24.9_{\pm0.1}$ | $41.2_{\pm0.3}$ |

[33], 2) *Modified objectives*: gradient matching with augmentation (DSA) [61], distribution matching (DM) [63], trajectory matching (MTT) [3], and flat trajectory distillation (FTD) [7].

**Setup** Following previous works, we employ standard ConvNet architectures [61, 9, 3] —three layers for $32 \times 32$ images and four layers for $64 \times 64$ images. We use the Higher package [17] to efficiently calculate the meta-gradients. We opt for a simple setup: using Adam for inner optimization with a learning rate of 0.001, and applying standard augmentations (flip and rotation) on the target set.

**Evaluation** For evaluations, we follow the protocol from [9, 61], assessing each distilled dataset on ten random neural networks and noting mean and standard deviation. For other baseline methods, we reference the best original paper results. Notably, [64, 33] uses a wider ConvNet to minimize surrogate approximation differences. Aligning with this, we conduct a transfer evaluation, distilling with a narrow network and evaluating with their wider one.

Further details, our code, and distilled checkpoints are in the Appendix.

### 3.1 Performance

Our simple approach demonstrates competitive performance across multiple datasets (Table 1). With 10 and 50 images per class, we achieve state-of-the-art results on the CIFAR-100, CIFAR-10, and CUB200 datasets without any inner loop approximations. Comparing all IPC values in $1, 10, 50$, our method matches the RCIG technique's performance across all datasets. Encouragingly, without

Figure 4: RaT-BPTT with parameterization.

| Dataset | CIFAR-10 | |
| Img/class(IPC) | 1 | 10 |
|---|---|---|
| **Parameterization** IDC [22] | $50.0_{\pm0.4}$ | $67.5_{\pm0.5}$ |
| LinBa [9] | $66.4_{\pm0.4}$ | $71.2_{\pm0.4}$ |
| HaBa [30] | $48.3_{\pm0.8}$ | $69.9_{\pm0.4}$ |
| Linear + RaT-BPTT | $68.2_{\pm0.4}$ | $72.8_{\pm0.4}$ |

optimizing specifically for wider networks, our approach still attains top results on CIFAR10, CIFAR100, and CUB200 for all IPC values. Evaluating datasets from broader-network methods on narrower configurations shows a performance decrease, e.g., a drop from 39.3% to 35.5% (RCIG, CIFAR100, IPC1) and from 25.4% to 22.4% (FrePO, TinyImageNet, IPC10). Our method seamlessly adapts to both wider and narrower networks, marking a distinct edge over previous benchmarks.

A separate and complimentary line of work aims to improve the optimization via parameterization of the distilled dataset [30, 22, 9]. Our method can be seamlessly combined with these techniques, and Fig. 4 shows the combination of linear parameterization [9] and RaT-BPTT. In configurations where parameterization surpasses standard RaT-BPTT, the combined method boosts performance by approximately 1.6%. We defer the ablation studies and discussion of limitations to the Appendix.

## 4 Conclusion

In this work, we proposed a simple yet effective method for dataset distillation, based on random truncated backpropagation through time. Through a careful analysis of BPTT, we show that randomizing the window allows to cover long dependencies in the inner problem while truncation addressed the unstable gradient and the computational burden. Our method achieves state of the art performance across multiple standard benchmarks, across both narrow as well as wide networks. We thus hope to provide a step towards model training at scale by advancing state-of-the-art of dataset distillation.

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

# A    Limitations

**Algorithm Design** The design of our method is primarily guided by intuitions and observations from empirical studies. Throughout the algorithm's development, we aim to strike a balance between scalability and effectiveness. Our approach currently involves tuning the ratio between the unrolling length and window size, scaling the unrolling length in accordance with the IPC number and GPU size. While this approach has demonstrated promise, we acknowledge that the current algorithmic choice might not represent the absolute optimal solution. Further research could investigate alternative algorithm designs.

**Application to larger models and datasets** A notable strength of our methodology is its versatility: it is compatible with all differentiable loss functions and network architectures, emphasizing its broad applicability. However, we only focus on illustrating the method's capabilities with standard benchmarks in the literature. This decision leaves a promising avenue for future work to apply and validate our method across various domains and tasks beyond image classification. It's also worth highlighting that while surrogate-based techniques are constrained to using the MSE loss to convexify the inner problem, our approach is agnostic to the specific loss function employed. This flexibility paves the way for our method's application in other realms, such as audio and text data distillation.

**GPU memory usage** Despite the significant improvements introduced by RaT-BPTT, it still necessitates unrolling and backpropagating over several steps, which require storing all intermediate parameters in the GPU. Consequently, this method incurs substantial memory consumption, often exceeding that of directly training the model. For larger models, one might need to implement checkpointing techniques to manage memory usage effectively.

# B    Related Work

Numerous follow up works have proposed clever strategies to improve upon the original direct bilevel optimization (Eq. (1)) in [54], like also learning soft labels [2, 48] (see [28, 41, 57, 15] for recent surveys and [6] for benchmarking). Most of them have focused on 1) approximating the function in the inner-loop with more tractable expressions, 2) changing the outer-loop objective and 3) re-parametrization of the data.

*Inner-loop surrogates:* The first innovative works [37, 38] tackle inner-loop intractability by approximating the inner network with the Neural Tangent Kernel (NTK) which describes the neural net in the infinite-width limit with suitable initialization ([19, 26, 1]) and allows for convex optimization, but scales unfavorably. To alleviate the scaling, random feature approximations have been proposed: [32] leverage a Neural Network Gaussian process (NNGP) to replace the NTK, using MC sampling to approximate the averaged GP. [64] propose to use the Gram matrix of the feature extractor as the kernel, equivalent to only training the last layer with MSE loss. A very recent work [33] assumes that the inner optimization is convex by considering linearized training in the lazy regime and replaces the meta-gradient with implicit gradients, thus achieving most recent state-of-the-art. Yet all of these approaches inevitably face the discrepancies between learning in the lazy regime and feature learning in data-adaptive neural nets (e.g. [16] and numerous follow ups) and often need to maintain a large model pool. Moreover, inner-loop surrogates, be it NTK, NNGP or random features, tend to show higher performance on *wide* networks, where the approximation holds better, and be less effective for the narrower models used in practice.

*Modified objective:* A great number of interesting works try to replace the elusive test accuracy objective with metrics that match the networks trained on full data and on synthetic data. [62] propose to match the gradient between the two networks with cosine similarity, with various variations (like differentiable data augmentation [61] (DSA)) and improvements ([20, 27]). Other works pioneer feature alignment [53], matching the training trajectories (MTT, introduced in [3] and refined in [12, 7, 60]), and loss-curvature matching [45]. More tangentially, note that the adaptation of optimization metrics has also been taken further to dataset generation for generalization attacks [58] or adversarial perturbations [49] as well as generating distilled data with a robustness objective [50].

*Data Parametrization:* A separate line of work aims to improve the optimization via parameterization of the distilled dataset. Since these works can be viewed as orthogonal to improving the bilevel optimization directly, we only mention them briefly: [30, 52] leverage encoder-decoder networks, [25, 4] use generative priors, [22, 31] propose multi-scale augmentation. Perhaps most relevant

to our work is [9] which gives a linear basis with weights for the dataset while remaining in the bilevel optimization framework directly. In principle, most of these ideas to parameterize the data can be combined with most of the methods to tackle the optimization in Eq. 1, so these constitute complimentary approaches to improve dataset distillation.

Dataset distillation shares many characteristics with *coreset selection* [21], which finds representative samples from the training set to still accurately represent the full dataset on downstream tasks. However, since dataset distillation generates synthetic samples, it is not limited to the set of images and labels given by the dataset and has the benefit of using continuous gradient-based optimization techniques rather than combinatorial methods, providing added flexibility and performance. The concept of coreset selection has been further refined within the realm of deep learning, whereby information is condensed for each forward pass, as elaborated by [10]. Both coresets and distilled datasets have found numerous applications including speeding up model-training [36], reducing catastrophic forgetting [43, 64], federated learning [18, 46] and neural architecture search [47].

## C  Experiments

### C.1  Experimental Details

**Data Preprocessing**: Leveraging a regularized ZCA transformation with a regularization strength of $\lambda = 0.1$ across all datasets, our approach adheres to the methods established by prior studies [37, 38, 64, 32, 9]. We apply the inverse ZCA transformation matrix for distillation visualization, using the mean and standard deviation to reverse-normalize optimized data.

**Models** Following previous works, we use Instance Normalization for all networks for both training and evaluation.

**Initialization** In contrast to conventional real initialization widely used in nearly all previous works, we employ random initialization for distilled data, hypothesizing that there is a reduction in bias from such uninformative initialization. Data are initialized via a Gaussian distribution and normalized to norm 1. For RaT-BPTT, we note comparable performance and convergence between random and real initialization.

**Label Learning** Following previous works that leverage learnable labels, we optimize both the data and label for CIFAR10-IPC50, all IPCs for CIFAR100, CUB-200, and Tiny-ImageNet. We forego normalization for label probability, hence the labels retain their positive real value representation.

**Training** In addition to the RaT-BPTT algorithm, we incorporate meta-gradient clipping with an exponential moving average to counter gradient explosion. We find that the proper combination of normalizing initialization and learning rate (0.001 for Adam) is crucial for successful distillation image training. While using instance normalization, an image scaled by $\alpha$ leads to meta-gradient scaling by $\frac{1}{\alpha}$. As a result, one should use an $\alpha$ times larger learning rate for Adam or $\alpha^2$ times larger for SGD to achieve the same optimization trajectory. We thus adopt a similar initialization scale to that of neural network training (normalized to norm 1), combined with a standard learning rate of 0.001 when using Adam. To maintain meta-gradient stability, we employ batch sizes of 5,000 for CIFAR-10 and CIFAR-100, 3,000 for CUB-200, and 1,000 for Tiny-ImageNet. Note that one should aim to further increase the batch size for Tiny-ImageNet until all the GPU memory is used.

**Hyperparameters** In an effort to minimize tuning requirements, we adhere to a standard baseline across all configurations. Specifically, we utilize the Adam optimizer for both the inner loop (network unrolling) and the outer loop (distilled dataset optimization) with learning rates uniformly set to 0.001. We refrain from applying weight decay or learning rate schedules that are used in prior works [64, 33].

**Evaluation** We evaluate our optimized data using a seperate held-out test dataset (the test set in the corresponding dataset). We adopt the same data augmentation as in previous work [9]. For depth 3 convolutional networks, we train using Adam with a learning rate of 0.001. No learning rate schedule is used.

### Code and Checkpoints

The code and checkpoints for RaT-BPTT could be found at `https://anonymous.4open.science/r/RaT-BPTT-45EE/`

## C.2 Ablations on the Random Truncated Window

In Section 2, we justify the necessity of performing truncation to speed up and stabilize the gradient, and the necessity of changing the truncated window to cover the entire trajectory. Now we provide an ablation study on how to select the truncated window. We compare three methods, 1) random uniform truncation, 2) backward moving, and 3) forward moving. For the forward (backward) moving method, we initialize the window at the end (beginning). It is then shifted forward (backward) by the window size whenever the loss remains stagnant for 2,000 steps.

From Figure 5, it is surprising that randomly uniform window placement achieves the best performance across the whole training process. A closer examination of the forward and backward moving curves suggests that altering the window's positioning can spur noticeable enhancements in accuracy. Such findings reinforce the idea that distinct truncation windows capture varied facets of knowledge, bolstering our intuition about the need for a comprehensive trajectory coverage by the window.

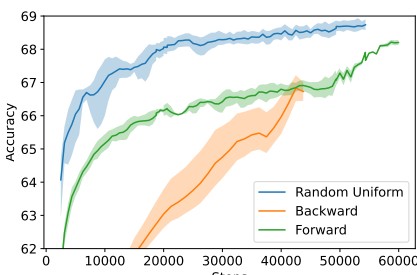

Figure 5: Comparison between random uniform truncation, backward moving, and forward moving. Random uniform truncation gives the best performance across the whole training process. N=120, T=40 for IPC10 with CIFAR10.

One might ask whether uniform sampling is the best design. Actually the answer is no. With careful tuning by sampling more on the initial phase, we find that one can further improve the final accuracy by 0.4% for CIFAR10 with IPC10. However, it introduces an additional hyper-parameter that requires careful tuning. To keep our method simple, we choose to go with the uniform one.

## C.3 Ablations on curriculum

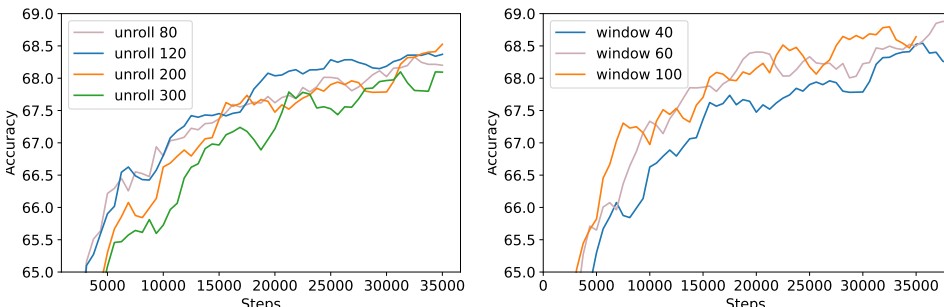

Figure 6: **Left** Test accuracy during distillation for different unrolling length of 80, 120, 200, 300 with fixed window size 40. CIFAR-10, IPC10. **Right** Test accuracy during distillation for different window size in 40, 60, 100 with fixed unrolling length 200.

Our RaT-BPTT implementation hinges on tuning two hyperparameters: unrolling length and backpropagation window size. This section presents an ablation study exploring these parameters for CIFAR-10, IPC10.

**Unroll length**

We initially fix the window size at 40 while varying the unrolling length. Notably, unrolling length governs the long-term dependencies we can capture within the inner loop. Fig 6 reveals that a moderate unrolling length, between twice and five times the window size, yields similar performance. However, disproportionate unrolling, as seen with a window size of 40 and unrolling length of 300, detrimentally affects performance.

**Window size**

Next, we fix the unrolling length at 200 and experiment with window sizes of 40, 60, and 100. Fig 6 shows the latter two sizes yield comparable performance. In RaT-BPTT, GPU memory consumption is directly proportional to the window size, thus a window size of 60, with an unrolling length of 200 steps, emerges as an optimal balance. As such, we typically maintain a window size to unrolling length ratio of around 1:3.

In our implementation, we employ a window size and unrolling length of (60, 200) for CIFAR-10 IPC1 and CUB-200, (80, 250) for CIFAR-10 IPC10, and (100, 300) for all other datasets.

### C.4 Other Metrics on Gradient Stability

In Figure 2, we investigated the stability of meta-gradients using gradient norms as a metric, predicated on the notion that stable and efficient learning should manifest as consistent and decreasing gradient norms throughout training. Expanding on this analysis, we now introduce another metric for evaluating gradient stability: the normalized gradient variance, in line with the methodology proposed by [13]. Each variance value reflects the instability across the batch samples, and the values across time steps reflects the instability across training steps.

To calculate this metric, we compute the average variance of all gradient entries using a set of 100 samples from the evaluation batch. Given the different scales in gradient norms across different methods, we normalize this variance against the square of the norm. This normalization yields a more consistent metric, termed the normalized variance. Employing the same experimental setup as in Figure 2, we present the results in Figure 7. It shows that RaT-BPTT not only maintains lower variance at each training step but also demonstrates more consistent variance trajectories over the course of training. These findings, in conjunction with the earlier results from Figure 2, collectively offer a comprehensive view of the argued training instability.

### C.5 Ablation on Stabilizing BPTT

In Figure 2, we have demonstrated the notable instability of meta-gradients via the gradient norm. This section extends our analysis with ablation studies, indicating that both controlling the gradient norm and incrementally increasing the unrolling parameter $T$ of BPTT result in only marginal improvements, which cannot compare to the gains garnered through RaT-BPP. We follow the setting in Figure 3.

Our foundational approach has already incorporated gradient clipping to manage extreme gradient norm values, employing a standard exponential moving average (EMA) with a $0.9$ decay rate and capping the gradient norm at twice the adaptive norm.

To further stabilize the gradient norm, we explored two additional methods: 1) BPTT-Gradient Clipping, limiting the gradient norm to no more than $1.05$ times the adaptive norm, and 2) BPTT-Normalized Gradient, ensuring a consistent gradient norm of 1 throughout training. However, as Figure 8 illustrates, these methods achieve only marginal enhancements over the basic BPTT approach. Their performance trails behind RaT-BPTT, with a threefold increase in optimization time due to extended backpropagation.

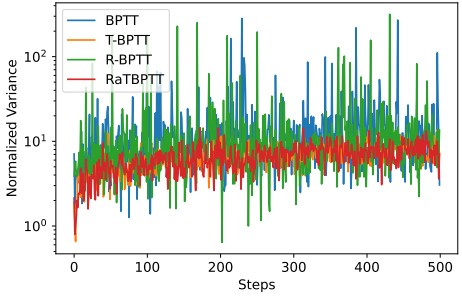
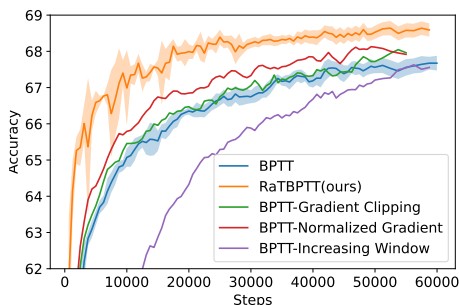

Figure 7: Normalized variance across batch samples of the meta-gradient. RaT-BPTT has stable and small variances. Same setting as in Figure 2.

Figure 8: Ablations on stabilizing BPTT: only controlling the gradient norm or gradually increasing the window is not enough.

These findings highlight challenges such as deviation from the kernel regime, variance from extensive unrolling, and pronounced non-convexity, contributing to gradient instability, as evidenced by fluctuating gradient norms. Addressing these issues solely by adjusting gradient norms proves insufficient.

Alternatively, we examine limiting the maximum Hessian matrices in Equation (2) by gradually extending the unrolling length $T$ in BPTT. In Figure 8, the BPTT-Increasing Windows variant, which linearly scales $T$ from 10 to 180, underperforms both R-BPTT and standard BPTT. This underlines the complexity within the inner loop, deviating significantly from the kernel regime and emphasizing the importance of managing the unrolling window size.

## C.6  Other Architectures

Table 2: Generalization to other architectures. We conduct transfer evaluation of a distilled dataset trained with a 3-layer ConvNet and directly training the dataset with the architecture in the inner loop. CIFAR10, IPC10.

| Architecture | VGG-11 | AlexNet | ResNet-18 |
|---|---|---|---|
| Transfer | $46.6_{\pm 0.9}$ | $60.1_{\pm 0.6}$ | $49.2_{\pm 0.8}$ |
| Direct Training | $47.7_{\pm 0.8}$ | $63.7_{\pm 0.6}$ | $53.0_{\pm 0.8}$ |

We further assessed our method across various architectures to demonstrate its universality. It is noteworthy that our approach is already effective across different widths of the convolutional networks (narrow and wide) we used. Additionally, we conducted tests using the standard VGG-11, AlexNet, and ResNet-18, both training it from scratch and transferring from the distilled dataset. To our knowledge, we are the pioneers in applying direct distillation to a standard-sized network like ResNet-18 and VGG-11. Prior works never train directly on VGG11 and they only use small or modified ResNets like ResNet-10 [59, 22], ResNet-12 [9] and ResNet-AP10 [22, 31] in these settings.

The results are presented in Table 2. Our results yield better or comparable transfer results compared with previous methods. Direct training further increases the numbers.

## C.7  Ablation on the Inner Optimizer

We have opted for Adam instead of SGD to simplify the tuning process for the inner loop. This decision was based on the ability to use a common learning rate without requiring decay in the inner loop. In this section, we perform ablation studies on how the inner loop optimizer affect the performance.

We implement RaT-BPTT (SGD) using SGD with learning rate 0.01 and learning rate decays at [120, 200] by 0.2. For IPC10 on CIFAR10, RaT-BPTT (SGD) achieves a 69.0% accuracy (std 0.3%), while RaT-BPTT (Adam) results in a slightly higher accuracy of 69.4% (std 0.4%). Thus, RaT-BPTT (SGD) also outperforms previous methods in this setting by a large margin. It is crucial to note that our improvement is attributed to factors beyond merely employing Adam in the inner loop.

It is also noteworthy to point out that we are not the first to use Adam for the inner loop during training. [32] also uses Adam for their linearized inner loop. Some other papers [33, 64] have also adopted Adam for the linear loop during evaluations. We suspect that whenever Adam was an option, the benchmarking papers probably tried it without significant improvements.

## C.8  Discussions on Efficiency

We have conducted a comparative analysis of the total training time for several methods, utilizing a consistent computational environment on an RTX8000 with 48GB. It should be noted that we have excluded RCIG from this comparison, as our reproduced accuracy falls short of the reported number. The following are the recorded training times (in hours) for CIFAR10 with IPC10: KIP (over 150), LinBa (100), FrePO (8), RaT-BPTT (22), MTT (14), and IDC (35). Among these methods, our cost ranks as the third best.

There are ways to further improve the efficiency. 1) The current package we utilize for meta-gradient calculation, the higher package, as noted in [35], lacks efficiency compared to other methods. We could lower the time cost by altering our implementation to more efficient methodologies. 2) The references [35, 40] contain efficient designs for the meta-gradient calculation. As reported in [40], it could lead to up to 2x speedups compared with the higher package. This improvement would not only enhance the performance of our method but also bring it in line with the efficiency benchmarks set by methodologies like FrePO. 3) Similar to FrePO, we may keep a pool of parameter checkpoints to further optimize our method. This strategy would reduce the need for inner training from new random initializations.

### C.9  Visualization

We incorporate visualizations for IPC10 on CIFAR-10, representing standardly trained (Figure 9), weakly boosted (Figure 11), and strongly boosted images (Figure 10). Upon inspection, the images from both boosted categories appear more diverse compared to their standard counterparts.

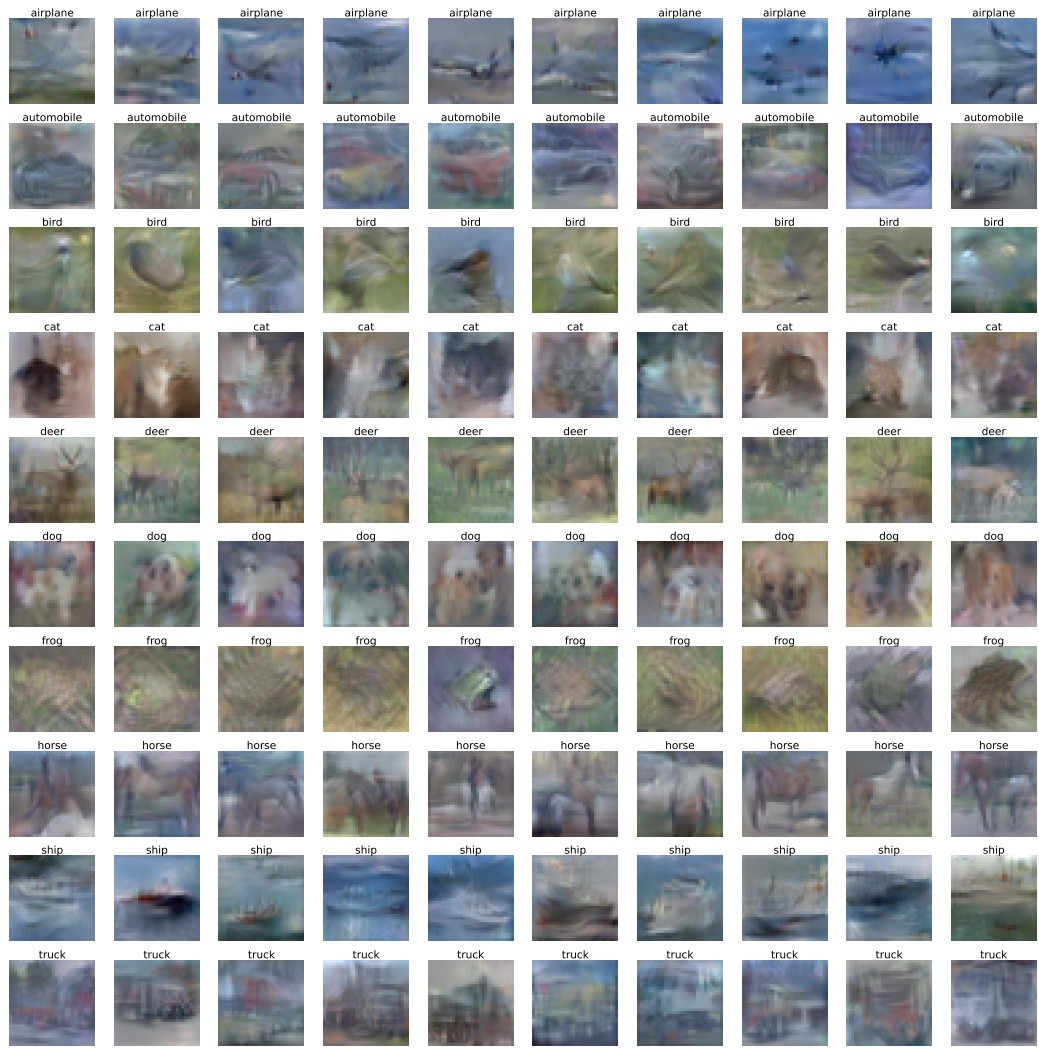

Figure 9: Visualization for RaT-BPTT standardly trained on CIFAR-10 with IPC10.

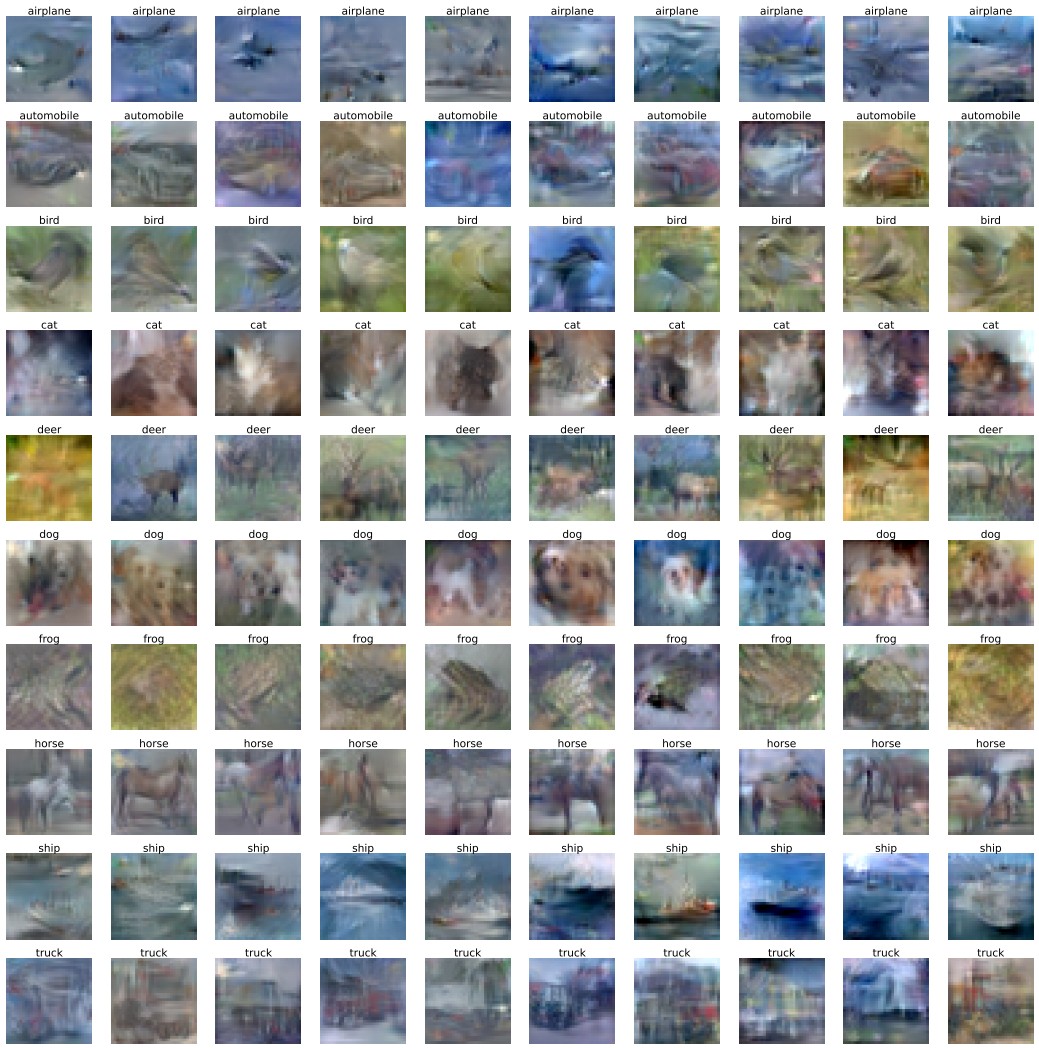

Figure 10: Visualization for RaT-BPTT with strong boosting (Boost-DD). CIFAR-10 with IPC10.

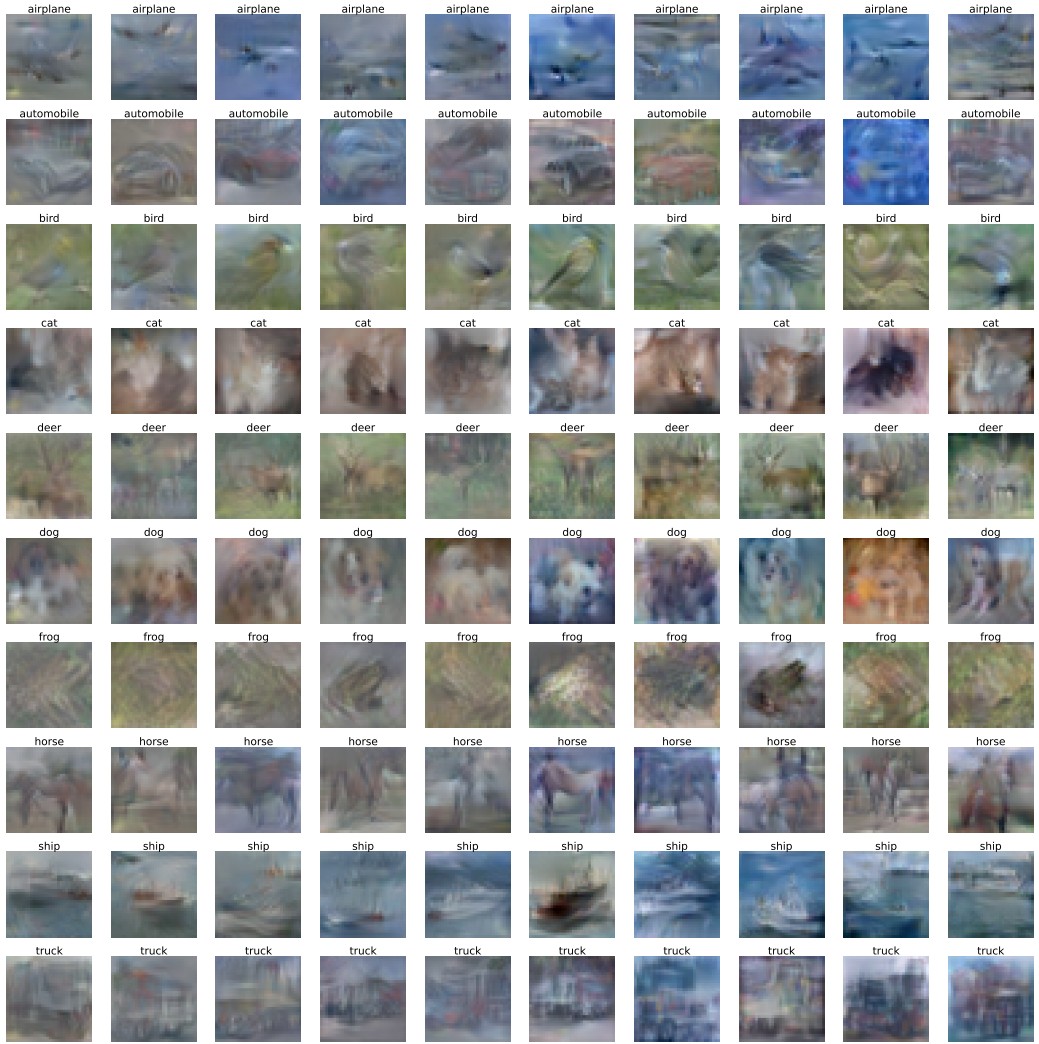

Figure 11: Visualization for RaT-BPTT with weak boosting (Boost-DD). CIFAR-10 with IPC10.

