# OpenReview forum: "Embarrassingly Simple Dataset Distillation"
_NeurIPS.cc/2023/Workshop/WANT — WANT@NeurIPS 2023 Poster_

### Official Review · Reviewer_2aH8 · 2023-10-23
**Good submission but some parts may be improved**

**Confidence:** 3

**Review:**

`Summary and contributions`

The authors introduces a new backpropogation algorithm "Random Truncated Backpropagation Through Time" (RaT-BPTT) to improve the dataset distillation process. The authors claims to achieve state-of-the-art for dataset distillation. The authors motivation was complex nature of back propagation and it's huge optimization and memory demands with Dataset distillation. They introduced randomization and truncation with BPTT used in dataset distillation (a bilevel optimization problem). The authors specify that, meta gradients for BPTT has greater storage requirement becuase of the need to store meta gradients for segments. They remove this requirement by adopting a window mechanism (i.e backpropagating by window of some steps). The method is tested on cifar-10, cifar-100, cub200, and Tiny Imagenet

`Strengths`

1. The paper introduces a novel method for improving the inner loop optimization in dataset distillation
2. It shows some signs of improvement over previous methods and efforts
3. The method is tested with good number of datasets robustly showing that it works for the given setup

`Weaknesses`

1. The paper does not show how the newly proposed method of optimization works with different networks for eg. bigger ResNets, Vision Transformer in case of computer vision. Without evaluation on bigger networks which has a big impact on long term dependencies, it's difficult to assert how well the method competes with BPTT.
2. As this is an improvement to optimization method, the paper should talk about it's application in different domains like vision, nlp etc and how the method can scale for larger networks
3. More

`Correctness`

The experimentation and results show good evidence that proposed method works appropriately on given datasets and architectures.

`Clarity`

The paper is clear enough but it should be a long pape rather than a short paper, as more details could be explained with related work and methodology.

`Reproducibility`

Most of the hyper-parameters are present by some are missing like degree of rotation, and amount of flipping. Also, hyper-parameters for Adam are missing. Setup section could be better and should provide more details for the approach to be fully reproducible.

---

### Official Review · Reviewer_CPCN · 2023-10-25
**Simple and elegant idea to improve dataset distillation**

**Confidence:** 3

**Review:**

Authors propose to randomize the trajectory window in truncated BPTT for dataset distillation. They show that doing so helps to better cover long-term dependencies in the optimisation problem and stabilizes gradients. The experiments on tiny image datasets prove the efficiency of the proposed approach.

The simplicity and meaningful performance improvements makes it valuable addition to dataset distillation research. As the proposed method is also computationally cheaper, it would be interesting to see its application to problems with more complex data, such as NLP.

---

### Official Review · Reviewer_8Lt9 · 2023-10-26
**interesting work for dataset distillation through bilevel optimization**

**Confidence:** 4

**Review:**

**Summary of the work**

This paper studies how to extract a small set of synthetic training samples from a large dataset with the goal of achieving competitive performance on test data when trained on this sample, thus reducing both dataset size and training time. This papers tackles dataset distillation by formulating it directly as a bilevel optimization problem, and proposes Random Truncated Backpropagation Through Time (RaT-BPTT) to stabilize the gradients and speeding up the optimization while covering long dependencies.

**Strength**

The studied framework is fundamental, and the proposed algorithm is efficient. Specially, the paper proposes an efficient dataset distillation by formulating it mathematically as a bilevel optimization problem, and proposes Random Truncated Backpropagation Through Time (RaT-BPTT) to stabilize the gradients and speeding up the optimization while covering long dependencies.

The results look promising. The proposed algorithm establish new dataset distillation state-of-the-art for several standard dataset benchmarks.

**Weakness**

The paper lacks review of related paper. For example, https://arxiv.org/pdf/2104.13114.pdf also formulates the data distillation problem (this paper calls it data subsampling) as bilevel optimization problem. It also proposes efficient approximation algorithm to solve the intractable non-convex problem. I'd love to see the authors discussed more about these papers.

---

### Meta-Review · Area_Chair_aYDE · 2023-10-27

**Recommendation:** Accept (Poster)
**Confidence:** 3

**Metareview:**

Interesting and elegant idea with a clear exposure. Results are promising and the only weakness seems to be that the paper could benefit from more details and a better discussion of related work. I recommend this work for a Poster.

---

### Decision · Program_Chairs · 2023-10-28

**Decision:**

Accept (Poster)

**Comment:**

We thank the authors for their time and contribution to WANT and we are pleased to share that after the reviewing process the paper has been accepted. Congratulations! We encourage the authors to consider reviewers' feedback for the improvement of the camera-ready version. We hope to see you in person at the workshop and brainstorm on efficient training research together!